# Carbon in Mineralised Plutons

**Joseph G. T. Armstrong** [1,*] **, John Parnell** [1] **and Adrian J. Boyce** [2]

1   School of Geosciences, University of Aberdeen, Meston Building, Aberdeen AB24 3UE, UK;
    j.parnell@abdn.ac.uk
2   Scottish Universities Environmental Research Centre (SUERC), University of Glasgow, Rankine Avenue,
    East Kilbride G75 0QF, UK; adrian.boyce@glasgow.ac.uk
*   Correspondence: joseph.armstrong@abdn.ac.uk

**Abstract:** The Paleoproterozoic schists of the Leverburgh Belt, South Harris and the Neoproterozoic carbonaceous metasediments of the Dalradian Supergroup were deposited during the two most significant periods of black shale deposition globally. Hosted within these metasedimentary rocks are graphite-bearing mineralised plutons, formed during orogenic events. The assimilation of carbonaceous lithologies during magmatic pluton emplacement is a commonly recognised mechanism in the formation of many metal and semi-metal-enriched deposits. Graphite mineralisation as a result of carbon assimilation is a feature often associated with these mineral deposits, though the source of the carbon and any associated metal deposits is not always understood. In this study, carbon and sulphur isotope analyses demonstrate that the crustal assimilation of the Paleoproterozoic host rocks took place during magmatic emplacement and provided the source of carbon and sulphur during mineralisation of the plutons. Minor enrichments of trace elements are present in the South Harris plutonic lithologies, indicating that mobilisation and enrichment occurred during assimilation of the schists. Petrographic and elemental analysis of a Dalradian-hosted Ordovician pluton indicates a similar but more substantial enrichment of these trace elements during crustal assimilation. The timing and depth of assimilation appear to play key roles in the extent of graphite and associated trace element enrichments.

**Keywords:** magmatic assimilation; graphite; mineralisation; plutons; base metals; sulphides; Paleoproterozoic; Dalradian; metasediments; elemental mobility

## 1. Introduction

Black shales and their metamorphic derivatives contain elevated concentrations of many redox sensitive elements relative to upper crustal averages, including copper (Cu), gold (Au), cobalt (Co), lead (Pb), manganese (Mn), molybdenum (Mo), vanadium (V), selenium (Se) and tellurium (Te) [1–3], due to their anaerobic depositional conditions and commonly sulphidic nature [4,5]. While the elemental concentrations of black shales are rarely of economic interest, their large lateral extent and depositional thicknesses make them a significant crustal sink of many of these important elements. The magmatic assimilation of a carbonaceous continental crust is a process by which these large sinks of economically important metals and semi-metals can become concentrated into enriched deposits [6]. Models of metal enrichment have been proposed for several economic deposits globally, including Duluth, Bushveld Complex, Aguablanca and Vammala [7–12]. The assimilation of carbonaceous country rock commonly results in graphite crystallisation within magmatic deposits and can provide evidence of elemental mobility from crustal sources [13]. Understanding the processes that control elemental enrichment and mobility within the crust is important for predicting and identifying regions of economic interest, particularly for metals and semi-metals with low average crustal abundance and increasing demand.

The depositional thickness and preservation of black shales has varied throughout geological time, dependent on the changing atmospheric and geographic conditions [2,5,14]. The data indicate two black shale depositional maxima have occurred, the largest at 1.9 to 2.0 Ga (Paleoproterozoic) and another at 0.5 to 0.6 Ga (late Neoproterozoic) (Figure 1), [15]. Post-deposition and lithification, many black shale deposits from these periods were encompassed within accretionary margins and subsequent orogenic events. Specific examples in the British Isles include the Paleoproterozoic graphitic schists of the Outer Hebrides and the Dalradian pelites of Scotland and Ireland. The assimilation of this carbonaceous country rock during orogenic intrusive activity may have provided a pathway for metal transport and enrichment for the formation of ore deposits [16].

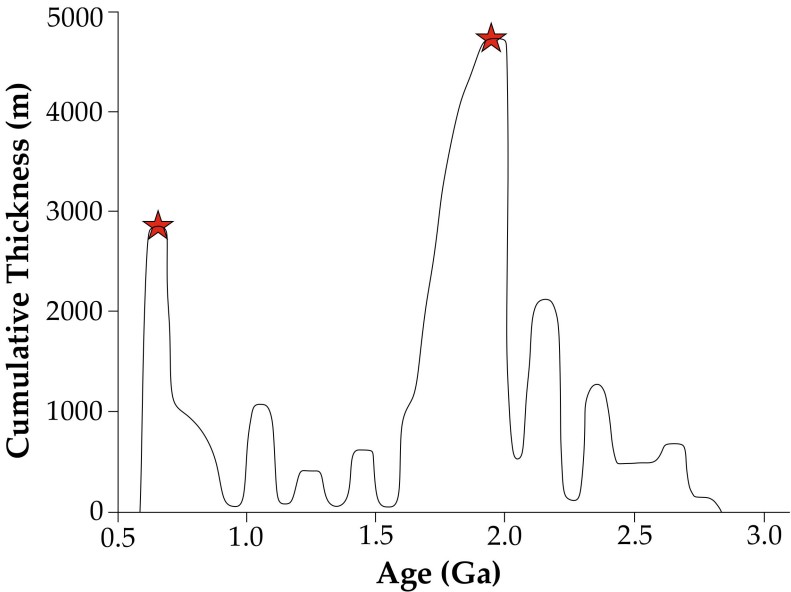

**Figure 1.** Cross-plot of cumulative thickness (m) vs. age (Ga) data for global shale deposition (after [15]). Relevant periods of shale depositional maxima marked with red stars, coinciding with the Paleoproterozoic Outer Hebrides schist (1.9 Ga) and Dalradian Supergroup (800–595 Ma) depositional ages.

This study will assess the potential for elemental enrichment during magmatic assimilation by utilising case studies from the Proterozoic black shale deposits of the Outer Hebrides (Paleoproterozoic schists) and the Scottish Highlands (Neoproterozoic pelites). Graphite-bearing intrusive suites hosted within these strata provide an opportunity to assess the evidence for the assimilation of carbonaceous country rock and the extent of elemental mobility and enrichment. Comparison of the carbon and sulphur isotope ratios from minerals in the graphitic intrusions and the adjacent country rocks may determine the presence and source(s) of assimilated graphite and sulphides in the intrusions.

*Geological Setting*

The Paleoproterozoic graphitic schists of the Outer Hebrides and the Neoproterozoic Dalradian pelites of Scotland and Ireland represent two key periods of black shale depositional maxima at 2.0 Ga and 0.5 to 0.6 Ga, respectively [15,17,18]. Given the increased abundance of redox-sensitive elements within black shales relative to the upper crustal averages, the extensive deposition and burial of black shales during these periods represented a potentially significant drawdown and accumulation of these elements from the surface environment into their respective basins [19–21]. These substantial black shale deposits were subsequently accreted during ocean closure events and underwent regional metamorphism during orogenesis [18,22]. Later phases of magmatic activity resulted in the emplacement of basic and ultrabasic intrusions within these carbonaceous host rocks, which contain abundant graphitised carbon. The assimilation of these black shales during

magmatic emplacement would have provided an abundant source of organic carbon for the recrystallisation of graphite in these intrusions [16].

The Paleoproterozoic schists of the Outer Hebrides form part of the Leverburgh Belt of the larger Harris Granulite Belt in South Harris (Figure 2A). Deposition of the Leverburgh Belt sediments occurred at c. 1.9 Ga, followed by accretion and crustal thickening of up to 35 km between 1910 and 1870 Ma, leading to granulite facies metamorphism of the sediments [18]. Subduction-related magmatism and heating occurred during this period of crustal thickening (1890–1888 Ma), with the emplacement of amphibolite facies of metanorite and metadiorite intrusions, which are commonly graphite-bearing.

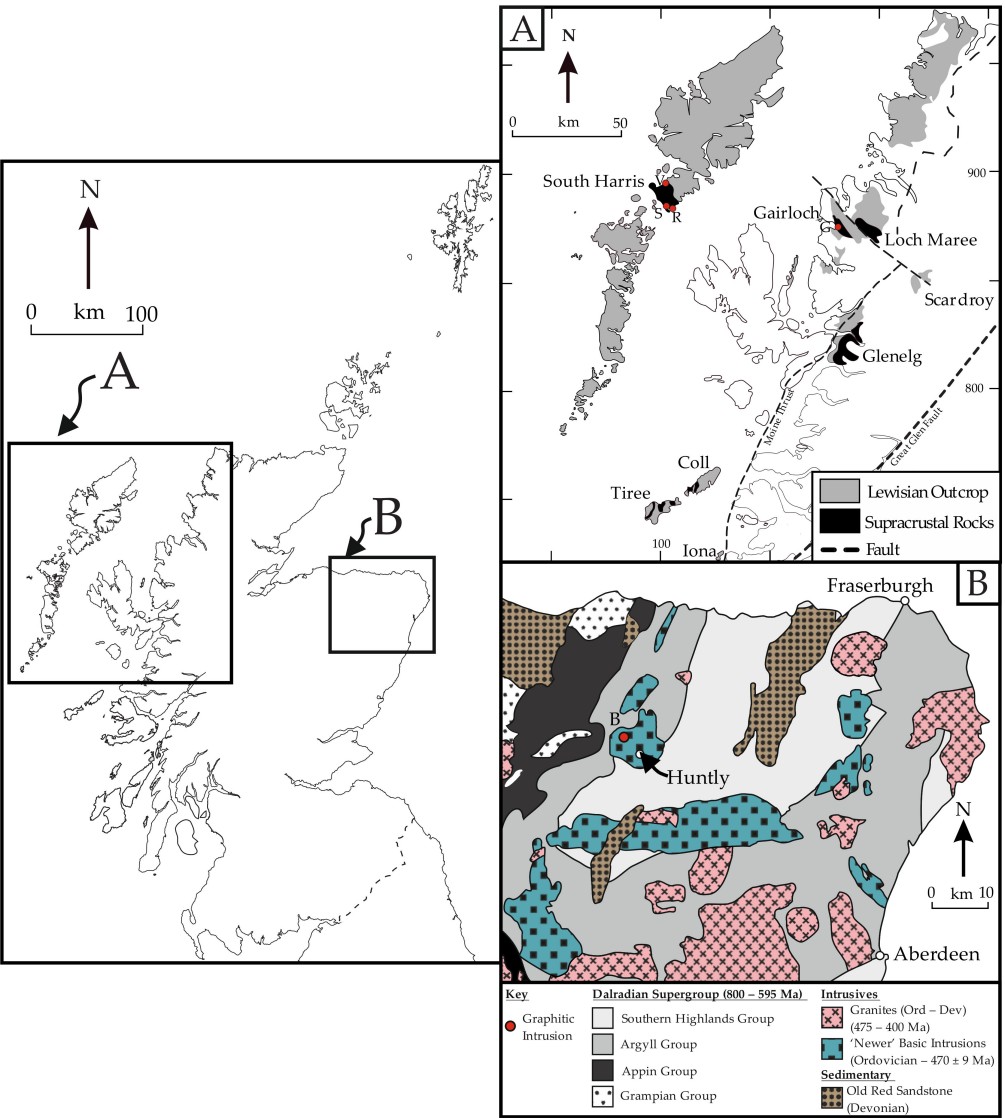

**Figure 2.** Regional geological maps of Scotland, with overview map for context. (**A**) Map of NW Scotland and Outer Hebrides (after [23]), illustrating the locations of Paleoproterozoic supracrustal rocks, with the study sampling locations highlighted in red: Stuaidh (S), Rodel (R), and Gairloch (G). (**B**) Map of NE Scotland (after [16,24]), illustrating the distribution of the Dalradian Supergroup metasedimentary units and basic and granitic intrusions in the region: Bin Quarry (B).

The Neoproterozoic metasedimentary Dalradian Supergroup in Northeast Scotland forms a significant portion of the Grampian Highlands terrane basement and outcrops across the region (Figure 2B). These metasediments were deposited in varying marine basinal conditions during 800–595 Ma [17,25], forming a sequence of sandstones, siltstones, carbonaceous shales and carbonates, with shallow marine glacial diamictites and volcanics

forming key correlation horizons (Figure 3). The granites and basic–ultrabasic layered intrusions of the Grampian Terrane formed during the Caledonian orogeny (475–400 Ma) [26], intruding and partially melting much of the Dalradian Supergroup sediments. During this period of crustal thickening, the Dalradian sediments also underwent widespread regional metamorphism and folding.

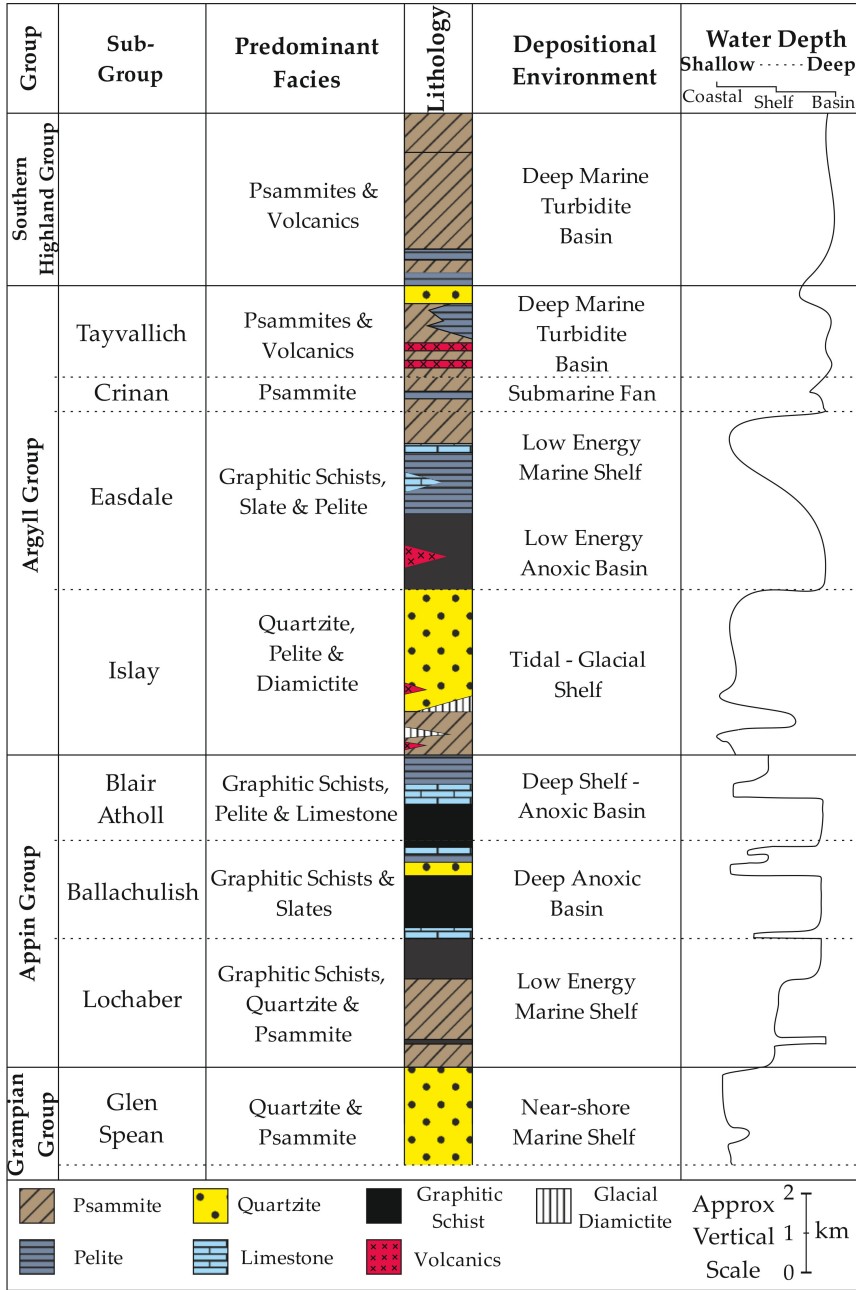

**Figure 3.** Simplified Dalradian Supergroup stratigraphy in NE Scotland (after [22,24,27]).

## 2. Materials and Methods

Appropriate outcrop samples with minimal weathering were collected from South Harris and Northeast Scotland and prepared at the University of Aberdeen for bulk elemental analysis and petrography. Bulk samples were crushed using a tungsten TEMA mill to a fine, homogenous powder. Representative petrographic samples were prepared by cutting with a diamond rock saw and fine polished using alumina powder for Scanning Electron Microscopy (SEM) analysis.

### 2.1. Whole-Rock Analysis

A bulk elemental analysis was performed on crushed samples of both graphitic and non-graphitic intrusions at ALS Labs, Loughrea, Ireland, using ALS methods ME-MS61L and ME-MS41L. ME-MS61L is a complete four-acid digest method, followed by neutralisation and analysis using ICP-MS and ICP-AES. ME-MS41L is an aqua regia partial-digest method, in which graphite, sulphides and most rock-forming minerals are fully digested, but some silicates may remain as a residue. Samples are neutralised before being analysed using ICP-MS and ICP-AES. Elemental concentrations are calculated based on the sample weight before digestion, which accounts for any undigested fractions.

A whole-rock total organic carbon (TOC) and total sulphur (S) analysis was performed on crushed samples at the University of Aberdeen using a LECO CS744 instrument. S concentrations were identified by a combustion analysis of an accurately weighed (~0.1 mg) powdered sample in the presence of excess tungsten and iron chip accelerators. TOC concentrations were determined by accurately weighing ~0.2 mg of powdered sample into a LECO standard filtering crucible. Samples were then digested using 25% HCl to remove any carbonate and subsequently neutralised using deionised water before air drying for 48-h. These decarbonated samples were then analysed using a LECO combustion analysis in the presence of excess tungsten and iron chip accelerators. Certified standards were utilised to produce a multipoint calibration before sample analyses and to check the calibration accuracy during analysis. Background C and S contents were accounted for by analysing blanks. Samples were analysed in duplicate to ensure a relative standard deviation (RSD) of <5%. Samples were reanalysed where appropriate. Data for TOC and S were given in wt.% relative to the (nondigested) whole rock.

### 2.2. Isotopic Analysis

Carbon and sulphur isotopic analyses were performed on separate samples of graphite and sulphides, respectively, at the Scottish Universities Environmental Research Centre (SUERC), East Kilbride. Samples were extracted from the bulk rock using a diamond-tipped drill and ground to a fine, homogenous powder using a mortar and pestle.

A carbon isotope analysis of graphite was performed by the following method. Samples of graphite were digested in 25% HCl over a 24-h period to remove trace carbonate and then neutralised using deionised water before air drying. Samples were weighed to the nearest μg using a high-accuracy balance into foil capsules and sealed. Samples were then analysed using a standard closed-tube combustion method under high vacuum by heating to 800 °C in the presence of 2 g of wire-form CuO over a 12-h period. Standards were utilised for analytical calibration with a reproducibility of <0.2‰. Data were denoted as per mille (‰) difference relative to the industry standard Vienna Pee Dee Belemnite (V-PDB).

Samples of sulphides for sulphur isotope analysis were accurately weighed and heated to combustion in an excess of copper oxide inside a high-vacuum line to oxidise all available sulphur into sulphur dioxide ($SO_2$). Utilising cold fingers and pentane traps, the $SO_2$ was separated from all extraneous gasses before analysis for the $^{34}S$ and $^{32}S$ isotope concentrations using a VG Isotech SIRA II mass spectrometer. Standards were utilised for the calibration of the mass spectrometer and testing accuracy, with a reproducibility of <0.2‰. Data were denoted as per mille (‰) difference relative to the industry standard Vienna-Canyon Diablo Troilite (V-CDT).

### 2.3. Scanning Electron Microscopy (SEM)

High-resolution SEM analyses were performed on polished sample blocks with a 10 nm C-coating, using a Zeiss Gemini SEM 300 Field Emission Gun Scanning Electron Microscope (FEG-SEM) at the University of Aberdeen Centre for Electron Microscopy, Analysis and Characterisation (ACEMAC). Analytical conditions: 20 kV accelerating voltage, 10.5 mm working distance and an aperture size of 60 μm. A Zeiss solid-state BSE detector (4-quadrant) was utilised for backscatter detection and imaging of mean atomic mass variations in samples. An Oxford Instruments X-Max 80 detector combined with AZtec

spectra interpretation software was used to conduct an electron dispersive X-ray (EDX) analysis to determine the elemental compositions at points across each sample. Elemental concentrations higher than ~0.5% are detectable using this method for elements with an atomic number of 6 or higher. Inter-elemental spectral interferences were accounted for during analysis.

## 3. Results

### 3.1. Field Observations and Sampling

The distribution of graphite and sulphides within the intrusive bodies at South Harris and Bin Quarry was observed to be heterogeneous at the outcrop scale. At South Harris, graphite and sulphides were identified in most exposures, though their abundance varied between the outcrops and sampling sites. Representative samples were collected, though exposures were predominantly found near the intrusion margin, so the distribution of graphite and sulphides across the intrusive suite was unclear. At Bin Quarry, the graphitic pyroxenite deposits occur as discrete, subvertical shear zones (Figure 4), within a wider intrusive body composed of non-graphitic, mafic and ultramafic, layered olivine gabbros and serpentinites [28]. Samples of both the graphitic pyroxenites and the associated layered olive gabbro at Bin Quarry were collected.

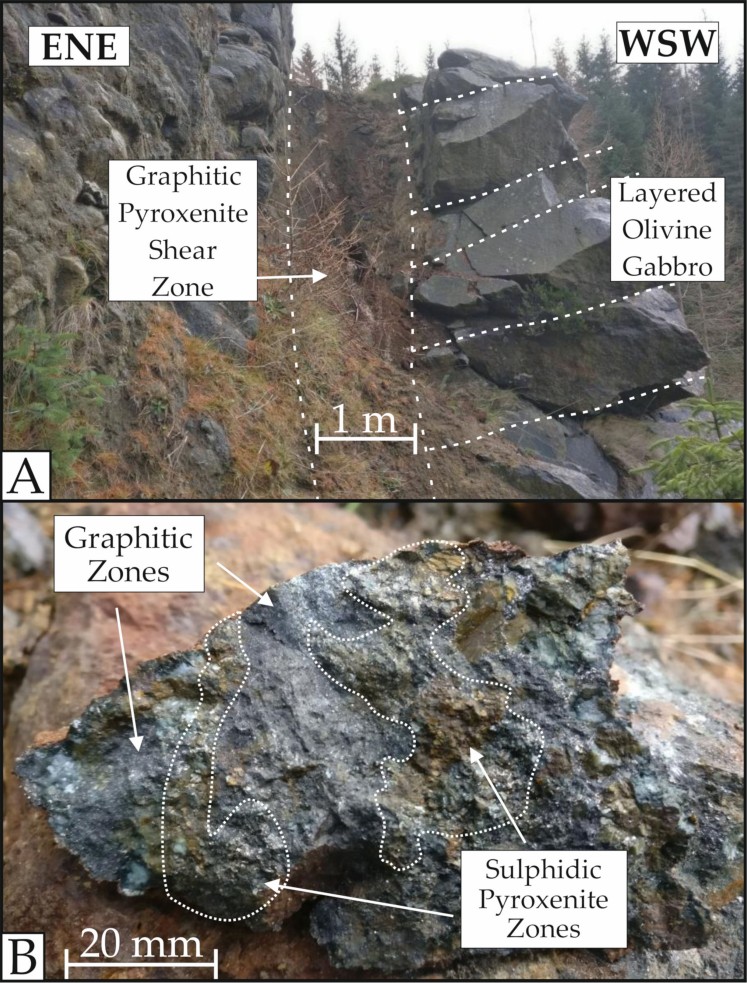

**Figure 4.** Annotated field photos from Bin Quarry, Northeast Scotland. (**A**) Outcrop photo showing a discrete, subvertical, graphitic pyroxenite shear zone (approx. 1 m thick) cross-cutting non-graphitic, layered olivine gabbro. (**B**) Hand specimen photo showing graphite and sulphide-rich zones within the graphitic pyroxenite lithology.

### 3.2. Scanning Electron Microscopy (SEM)

The mineralogy of the South Harris graphitic intrusions is variable, with two distinct end-member lithologies present in the samples analysed. One lithology is predominantly felsic in mineral composition (CAT11), and the other is predominantly mafic in composition (CAT12). The typical composition of the felsic (metagranite) lithology (Figure 5A) is biotite (25%); plagioclase (20%); K-feldspar (20%); quartz (15%); chlorite (5%); garnet (5%); graphite (5%); pyrite (5%) and accessory minerals, including chalcopyrite, molybdenite and jarosite (<1%). The typical composition of the mafic (metagabbro) lithology is (Figure 5B) clinopyroxene (25%); almandine garnet (20%); amphibole (15%); chlorite (10%); plagioclase (10%); epidote (5%); muscovite (5%); graphite (5%); pyrite (5%) and accessory minerals, including Ti-oxide (rutile), ilmenite, sphene, pyrrhotite and chalcopyrite (<1%). In both lithotypes, graphite occurs as discrete, elongate, lath-shaped minerals 0.1–0.8 mm in length. Pyrite and other sulphides commonly occur as discrete euhedral minerals (0.05–0.5 mm) and, occasionally, as cross-cutting sulphide veins (10–20 μm wide). Graphite and sulphides are typically larger in the metagabbro than in the metagranite.

A petrographic analysis of samples from the Bin Quarry ultrabasic intrusion in Northeast Scotland confirmed the presence of abundant graphite and sulphides in these deposits (Figure 6). The morphology of the graphite in the intrusion is comparable to the lath-shaped graphite in the South Harris intrusives, though it is more abundant at Bin Quarry and commonly occurs as smaller crystals (0.1–0.3 mm). The Bin Quarry intrusion contains abundant bytownite (plagioclase) phenocrysts (3–5 mm width), as previously identified in other works [29], in which the sample was described as an ultrabasic graphitic pyroxenitic pegmatite. Graphite and sulphides (pyrrhotite) within the Bin Quarry intrusion are present within both the bytownite phenocrysts and the wider groundmass. The modal mineralogy of the Bin Quarry graphitic pyroxenite is as follows: pyrrhotite (30%); graphite (30%); bytownite phenocrysts (20%); augite phenocrysts (5%); pyroxene groundmass (5%); albite (5%); pentlandite (3%); orthoclase (2%) and accessory minerals, including pyrite, chalcopyrite, Ti-oxide (rutile) and quartz (<1%).

### 3.3. Whole-Rock Analysis

Bulk elemental analyses of the South Harris graphitic metagabbro and metagranite (Table 1) show minor enrichments in copper (Cu), manganese (Mn) selenium (Se), tellurium (Te), tungsten (W) and zinc (Zn) relative to the average upper crustal values [30–32]. Whole-rock analyses of the Bin Quarry graphitic pyroxenite (Table 1) demonstrates significant enrichments in cobalt (Co), Cu, nickel (Ni), Se, Te and W relative to the upper crustal values, with minor enrichments in silver (Ag), molybdenum (Mo) and palladium (Pd). Comparisons with elemental concentrations in the non-graphitic olivine gabbro at Bin Quarry (Table 1) demonstrates that the graphitic pyroxenite is relatively enriched in gold (Au), Ag, Co, Cu, Mo, Ni, lead (Pb), Pd, Se and Te by at least an order of magnitude (10×). The majority of the enrichments seen in the graphitic pyroxenites are more significant than those found in the South Harris graphitic plutons (Figure 7).

The TOC and S analyses of the South Harris graphitic intrusions (Table 2) indicates a significant difference between the metagranite (CAT11) and the metagabbro (CAT12). The TOC values for the metagranite are 1.45–2.37%, with an average value of 1.80% ($n = 6$), while the TOC values obtained for the metagabbro are 0.11–0.75%, with an average of 0.32% ($n = 6$). The S concentration of the metagranite is also typically higher than that of the metagabbro, with metagranite S values of 0.23–0.44%, with an average of 0.33% ($n = 6$) compared with 0.01–0.09%, with an average of 0.03% ($n = 5$) in the metagabbro. These data omit one outlier S value of 1.75% (CAT12A). The presence of high S in one of the mafic intrusion samples likely indicates a vein of discrete sulphide mineralisation in this sample and is not representative of the bulk rock average. Sulphide veining was observed in the SEM petrographic analysis of this sample (Figure 5B).

**Table 1.** Whole-rock elemental analysis data of the South Harris and Bin Quarry graphitic intrusions (data sourced from [24]) and the Bin Quarry olivine gabbro. Upper crustal average values data from [30–32].

| Locality | Lithology | Lab Code | Au (ppm) | Ag (ppm) | Co (ppm) | Cr (ppm) | Cu (ppm) | Mn (ppm) | Mo (ppm) | Ni (ppm) | Pb (ppm) | Pd (ppm) | Pt (ppm) | Se (ppm) | Te (ppm) | Ti (%) | V (ppm) | W (ppm) | Zn (ppm) |
|---|---|---|---|---|---|---|---|---|---|---|---|---|---|---|---|---|---|---|---|
| South Harris | Meta-granite | CAT11 | 0.0033 | 0.119 | 21.3 | 116.5 | 98 | 843 | 4.64 | 12.2 | 8.15 | <0.002 | 0.006 | 1.33 | 0.07 | 0.709 | 193.5 | 142.5 | 99.4 |
| | Meta-gabbro | CAT12 | 0.0050 | 0.183 | 46 | 286 | 178.5 | 2570 | 4.18 | 82.2 | 3.76 | <0.002 | 0.004 | 1.96 | 0.152 | 1.195 | 374 | 148 | 186 |
| Bin Quarry | Graphitic Pyroxenite | STORM 708 | 0.0059 | 0.210 | 157 | 147 | 564 | 153.5 | 10.15 | 661 | 2.8 | 0.011 | 0.002 | 3.80 | 0.34 | 0.036 | 41.3 | 114 | 9.5 |
| | | STORM 708B | 0.0035 | 0.221 | 198 | 142 | 556 | 110 | 13.4 | 774 | 3.5 | 0.014 | 0.003 | 3.80 | 0.33 | 0.040 | 39.8 | 111.5 | 10.0 |
| | | STORM 709A | 0.0017 | 0.097 | 99.5 | 199 | 225 | 104 | 5.06 | 339 | 2.16 | 0.006 | 0.002 | 1.40 | 0.18 | 0.043 | 56.8 | 97.3 | 11.1 |
| | | STORM 709B | 0.0015 | 0.279 | 454 | 192 | 618 | 63.5 | 20.1 | 1370 | 3.27 | 0.033 | <0.002 | 6.00 | 0.95 | 0.039 | 52.9 | 137 | 11.1 |
| | | STORM 710A | 0.0016 | 0.368 | 320 | 137.5 | 1050 | 89 | 22.9 | 1500 | 4.27 | 0.026 | 0.003 | 8.90 | 1.26 | 0.027 | 41.2 | 107 | 6.3 |
| | | STORM 710B | 0.0023 | 0.304 | 345 | 165.5 | 1230 | 154 | 24.4 | 1610 | 2.17 | 0.043 | <0.002 | 8.60 | 0.99 | 0.030 | 48.9 | 42.4 | 11.3 |
| | | STORM 710C | 0.0025 | 0.344 | 358 | 165.5 | 776 | 97 | 25.8 | 1520 | 3.25 | 0.018 | <0.002 | 7.90 | 0.94 | 0.030 | 44.2 | 74.1 | 5.9 |
| | | STORM 710D | 0.0030 | 0.353 | 234 | 176.5 | 534 | 119 | 17.35 | 1010 | 2.26 | 0.013 | 0.003 | 5.40 | 0.68 | 0.034 | 47.2 | 42.8 | 7.5 |
| | | STORM 721 | <0.0002 | 1.080 | 1050 | 104.5 | 1050 | 26 | 74.8 | 5870 | 12.8 | 0.057 | 0.002 | 24.90 | 3.64 | 0.038 | 30.4 | 133.5 | 7.0 |
| | | STORM 722 | <0.0002 | 0.341 | 555 | 57.9 | 1245 | 196 | 44.9 | 1820 | 5.65 | 0.081 | 0.003 | 16.70 | 1.62 | 0.138 | 63.9 | 680 | 31.7 |
| | | Average (*n* = 10) | 0.0022 | 0.360 | 377 | 149 | 785 | 111 | 25.9 | 1647 | 4.2 | 0.030 | 0.003 | 8.74 | 1.09 | 0.046 | 46.7 | 154 | 11.1 |
| | | 1σ | 0.0014 | 0.244 | 269 | 54 | 383 | 692 | 19.3 | 1475 | 3.0 | 0.023 | 0.001 | 6.65 | 0.94 | 0.353 | 96.1 | 163 | 52.5 |
| | Olivine Gabbro | STORM 712A | 0.0002 | 0.011 | 39.5 | 46.8 | 28.5 | 186 | 0.09 | 45 | 0.167 | 0.001 | <0.002 | 0.1 | 0.02 | 0.011 | 11.3 | 105.5 | 12.9 |
| Global | Upper Crustal Average Values | | 0.0015 | 0.053 | 17.3 | 92 | 28 | 527 | 1.1 | 47 | 17 | 0.0005 | 0.0005 | 0.09 | 0.027 | 0.38 | 97 | 1.9 | 67 |

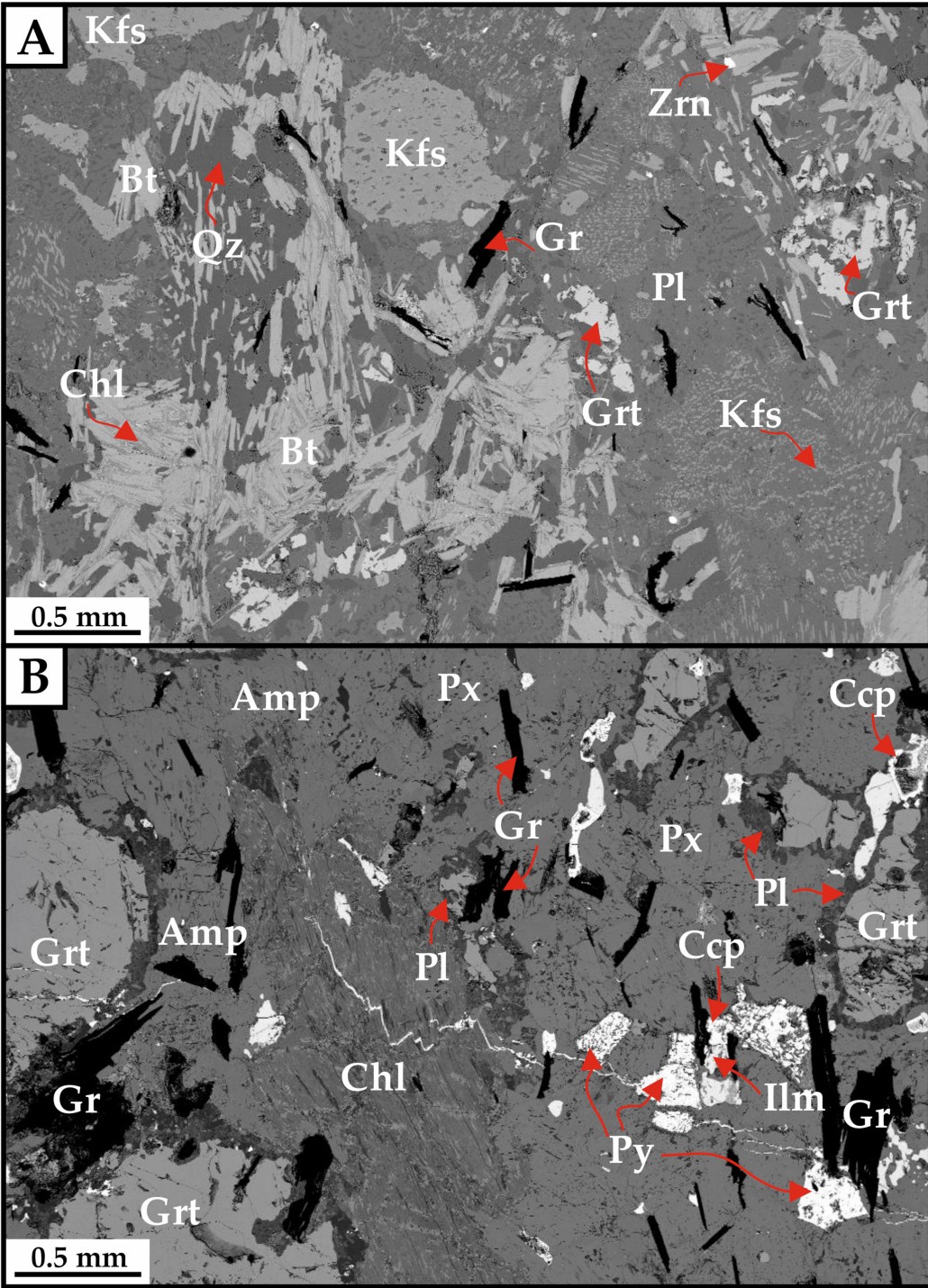

**Figure 5.** Scanning electron microscopy (SEM) backscatter imagery of the South Harris (Rodel) graphitic intrusions with the mineralogy labelled: (**A**) Metagranite; (**B**) Metagabbro; Amphibole (Amp); Biotite (Bt); Chlorite (Chl); Chalcopyrite (Ccp); K-Feldspar (Kfs); Garnet (Grt); Graphite (Gr); Ilmenite (Ilm); Plagioclase (Pl); Pyrite (Py); Pyroxene (Px); Zircon (Zrn). Mineral abbreviations as per IMA-CNMNC approved mineral symbols list [33].

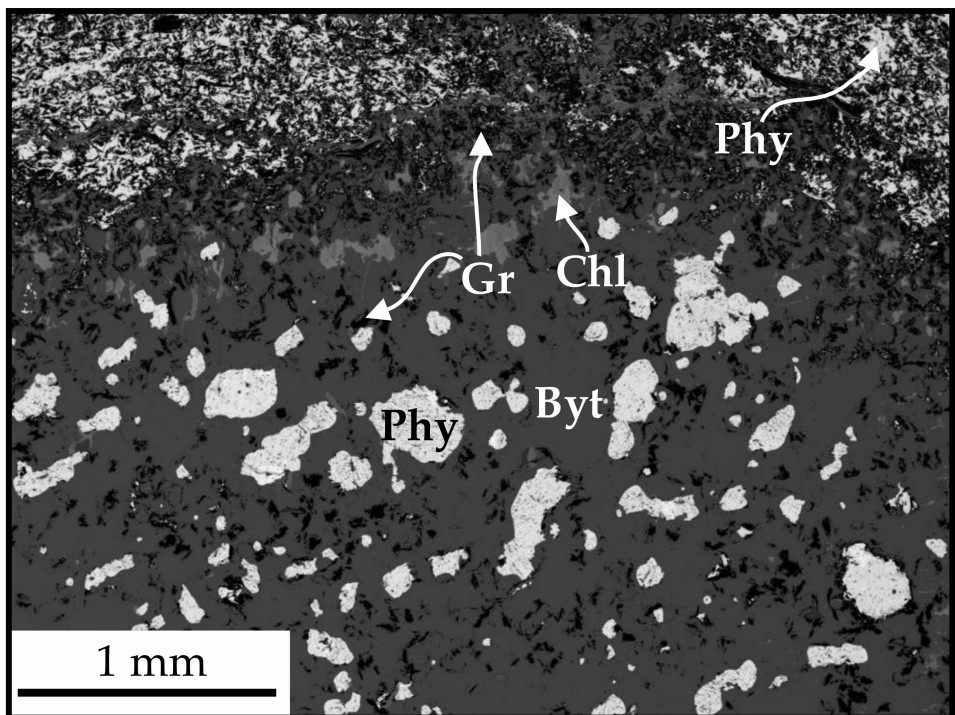

**Figure 6.** SEM backscatter image of the Bin Quarry graphitic pyroxenite intrusion, demonstrating the large bytownite phenocrysts and surrounding groundmass with abundant graphite and pyrrhotite: Bytownite (Byt); Chlorite (Chl); Graphite (Gr); Pyrrhotite (Phy). Mineral abbreviations as per IMA-CNMNC approved mineral symbols list [33].

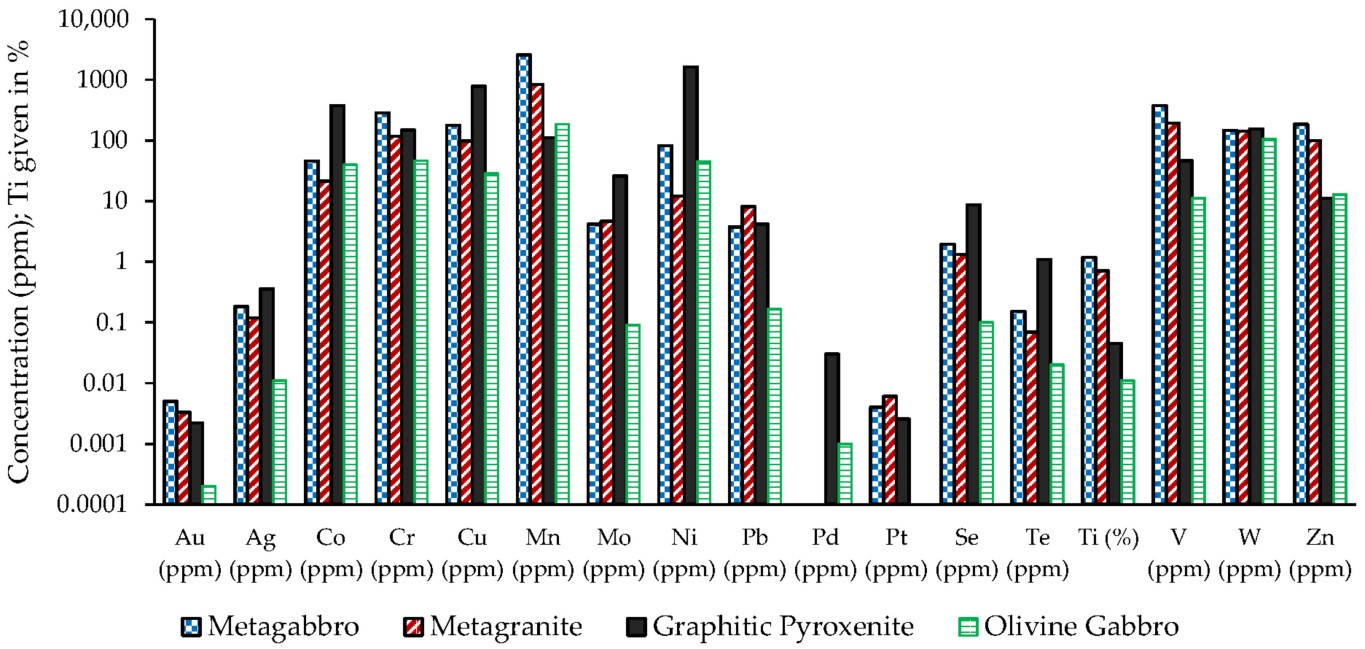

**Figure 7.** Bar chart of the whole-rock elemental concentrations for the South Harris metagabbro, metagranite, the average elemental concentrations (*n* = 10) of the Bin Quarry graphitic pyroxenite and the Bin Quarry olivine gabbro intrusion.

**Table 2.** Whole-rock TOC and S analysis data (LECO) of the South Harris graphitic intrusions. Bin quarry TOC and S data provided for comparison. Bin Quarry graphitic pyroxenite data sourced from [16].

| Locality | Lithology | Lab Code | TOC (%) | S (%) |
|---|---|---|---|---|
| South Harris | Metagranite | CAT11A | 1.60 | 0.26 |
| | | CAT11B | 1.45 | 0.43 |
| | | CAT11C | 1.90 | 0.44 |
| | | CAT11D | 2.37 | 0.32 |
| | | CAT11E | 1.66 | 0.23 |
| | | CAT11F | 1.81 | 0.30 |
| | | Average (*n* = 6) | 1.80 | 0.33 |
| | | 1σ | 0.29 | 0.08 |
| | Metagabbro | CAT12A | 0.54 | 1.75 |
| | | CAT12B | 0.75 | 0.01 |
| | | CAT12C | 0.19 | 0.02 |
| | | CAT12D | 0.11 | 0.09 |
| | | CAT12E | 0.14 | 0.02 |
| | | CAT12F | 0.18 | 0.01 |
| | | Average (*n* = 6) | 0.32 | 0.31 |
| | | 1σ | 0.24 | 0.64 |
| Bin Quarry | Graphitic Pyroxenite | STORM 708 | 0.86 | 2.70 |
| | | STORM 708B | 1.27 | 3.16 |
| | | STORM 709A | 0.84 | 1.32 |
| | | STORM 709B | 2.41 | 5.59 |
| | | STORM 710A | 6.05 | 6.31 |
| | | STORM 710B | 3.43 | 6.67 |
| | | STORM 710C | 4.30 | 6.59 |
| | | STORM 710D | 3.22 | 4.45 |
| | | STORM 721 | 15.05 | 18.60 |
| | | STORM 722 | 11.95 | 9.51 |
| | | Average (*n* = 10) | 4.94 | 6.49 |
| | | 1σ | 4.60 | 4.62 |
| | Olivine Gabbro | STORM 712A | 0.00 | 0.16 |

*3.4. Isotopic Analysis*

The C-isotope analysis of graphite within the South Harris felsic intrusions (Table 3) yielded $\delta^{13}C$ values of −22.3‰, −23.9‰ and −19.6‰. The typical mantle-derived carbon $\delta^{13}C$ values are −5 ± 3‰ [34], indicating that the carbon in these intrusions was not sourced from native magma. Comparable $\delta^{13}C$ values have been found in nearby graphitic schists [23] (Table 3).

The S-isotope analysis of pyrite from the metagranite found values of +0.0 and +0.1‰, while the $\delta^{34}S$ analysis of pyrite in the metagabbro returned values of +0.7 and +5.3‰ (Table 4). These values are slightly higher than the depleted mantle average $\delta^{34}S$ of −1.28 ± 0.33‰ [35], indicating a non-magmatic source for sulphide crystallisation or a mixed mantle plus country rock sulphur source. The $\delta^{34}S$ values for the proximal sulphide-bearing supracrustal lithologies analysed ranged from −1.3 to +16.3 (*n* = 8).

**Table 3.** Carbon isotope data on graphite from the South Harris intrusions. Data for proximal Paleoproterozoic carbonaceous country rocks were provided for comparison, data sourced from [23]. Carbon isotope data on graphite from the Bin Quarry intrusion and proximal carbonaceous country rocks were provided for comparison and discussion, data sourced from [16].

| Region | Context | Locality | Lithology | Lab Code | Grid Reference | $\delta^{13}$C |
|---|---|---|---|---|---|---|
| South Harris | Intrusion | Rodel | Metagranite | HAR1 | NG 052833 | −22.3 |
| | | | | HAR2 | NG 052833 | −23.9 |
| | | | | HAR3 | NG 052833 | −19.6 |
| | Country Rock | Rodel Pier | Schist | PPG13 | NG 047830 | −24.2 |
| | | Rodel Church | Schist | PPG15 | NG 048832 | −25.1 |
| | | | | PPG16 | NG 048832 | −24.9 |
| | | Stuaidh | Schist | PPG10 | NG 043832 | −24.5 |
| | | | | PPG11 | NG 043832 | −24.8 |
| | | | | PPG12 | NG 043832 | −25.0 |
| Wester Ross | Country Rock | Gairloch | Schist | PPG6 | NG 822736 | −24.0 |
| | | | | PPG7 | NG 822736 | −24.5 |
| | | | | PPG8 | NG 822736 | −24.4 |
| | | | | PPG9 | NG 822736 | −24.4 |
| | | | | PPG26 | NG 822736 | −23.6 |
| NE Scotland | Intrusion | Bin Quarry | Graphitic Pyroxenite | WAB11 | NJ 498430 | −21.8 |
| | | | | WAB12 | NJ 498430 | −21.2 |
| | Country Rock | Allt Nathrach | Amphibolite | WAB10 | NJ 1510 | −19.8 |
| | | Cairn of Claise | Pelite | WAB38 | NO 185795 | −23.0 |
| | | Glenbuchat | Pelite | WAB13 | NJ 336177 | −22.4 |
| | | Mortlach | Pelite | WAB34 | NJ 442482 | −26.1 |
| | | | | WAB35 | NJ 442482 | −18.5 |
| | | Portsoy | Pelite | AW1 | NJ 585685 | −26.5 |
| | | Altanower Forest | Pelite | WAB4 | NO 0882 | −16.4 |
| | | | | WAB5 | NO 0882 | −16.5 |
| | | Coulins Burn | Pelite | WAB16 | NJ 325187 | −22.9 |
| | | | | WAB17 | NJ 325187 | −23.8 |

**Table 4.** Sulphur isotope data on sulphides from the South Harris intrusions and proximal Paleoproterozoic sulphidic country rocks.

| Context | Locality | Lithology | Mineral | Lab Code | Grid Reference | $\delta^{34}$S |
|---|---|---|---|---|---|---|
| Intrusion | Rodel | Metagranite | Pyrite | CAT11-dS1 | NG 052833 | 0.0 |
| | | | | CAT11-dS2 | NG 052833 | 0.1 |
| | | Metagabbro | Pyrite | CAT12-dS1 | NG 052833 | 5.3 |
| | | | | CAT12-dS2 | NG 052833 | 0.7 |
| Country Rock | Rodel | Marble | Pyrrhotite | CAT32-dS1 | NG 048832 | 11.3 |
| | | | | CAT32-dS2 | NG 048832 | 11.4 |
| | | | | CAT32-dS3 | NG 048832 | 11.6 |
| | Langavat | Marble | Pyrite | CAT33-dS1 | NG 048832 | 10.2 |
| | Stuaidh | Graphitic Schist | Pyrite | JPDEEP181 | NG 042832 | 15.8 |
| | | | | JPDEEP182 | NG 042832 | 16.3 |
| | Borve | Sulphidic Gneiss | Pyrrhotite | JPDEEP 159 | NG 027949 | −1.3 |
| | | | | JPDEEP160 | NG 027949 | 2.0 |

## 4. Discussion

### 4.1. Evidence for Crustal Assimilation

The TOC concentrations in the South Harris plutons (Table 2) range from 0.11 to 2.37% (*n* = 12), indicating a significant, though variable, quantity of graphite within both deposits, which is supported by the identification of graphite using SEM and in-hand specimens. Notably, the metagranite has a higher average TOC concentration (1.80%) compared to the

metagabbro samples (0.32%). This carbon could be mantle-derived or derived through the assimilation of carbonaceous host lithologies. The C-isotope values ($\delta^{13}C$) of the graphite in the metagranites range from −23.9 to −19.6‰ (Table 3). This contrasts with a typical mantle $\delta^{13}C$ value of −5 ± 3‰ [34], indicating that the carbon in these intrusions is not mantle-derived. The proximal Paleoproterozoic carbonaceous schists of the Leverburgh Belt metasediments at Rodel and Stuaidh (Figure 2) are a potential carbon source, given the potential for partial melting of the host rock during magmatic emplacement. Comparable graphite $\delta^{13}C$ values of −24.2 to −25.1‰ [23] for these proximal lithologies support a model of carbon assimilation from the Leverburgh Belt metasediments. High TOC and S concentrations of up to 3.9% and 4.7%, respectively, in these proximal schists provided an abundant source of both carbon and sulphur for assimilation and recrystallisation as graphite and sulphides [23]. Abundant sulphide mineralisation is observed in the South Harris intrusive samples (Figure 5), which could have been sourced through partial melting and the assimilation of these sulphidic schists.

Sulphur isotope ($\delta^{34}S$) values for sulphides from the South Harris intrusions range from 0.0 to 5.3‰ (Table 4), which is slightly higher than the depleted mantle average of −1.28 ± 0.33‰ [35]. The $\delta^{34}S$ values for sulphides from the proximal Paleoproterozoic host rocks at Rodel, Stuaidh and Langavat (+10.2 to +16.3‰) are significantly higher than those obtained from the intrusive sulphides (Table 4), which suggests there may have been a mixed mantle + crustal sulphur source during sulphide crystallisation in the Harris intrusions. Alternatively, sulphidic gneiss at Borve (10 km north of Rodel) contains sulphides with $\delta^{34}S$ ratios of between +2.0 and −1.3‰, indicating that partial melting and assimilation of this lithology may have occurred at depth during South Harris magmatic emplacement. Comparable $\delta^{34}S$ ratios of −0.8 to +2.1‰ are also found in sulphides from the Kerry Road VMS deposit at Gairloch [36], within the Paleoproterozoic Loch Maree Group (c. 2.0 Ga) approximately 75 km west of the South Harris intrusions (Figure 2). It is feasible that such an abundant source of sulphides could have saturated the magma during assimilation, though the large distance between the Kerry Road VMS deposit and the site of magmatic crystallisation on South Harris makes this an improbable sulphur source.

Based on the $\delta^{13}C$ and $\delta^{34}S$ data from the South Harris plutons and surrounding lithologies, it is clear that magmatic assimilation has occurred during magmatic emplacement. These data suggest that carbon has been sourced from the Leverburgh belt metasediments, while the sulphur source is less definitive. Based on the $\delta^{34}S$ data, the assimilation of sulphides from the Borve sulphidic gneiss at depth is probable, though a mixed mantle and Leverburgh metasediment sulphur source is possible. Due to the high-grade metamorphic history of these deposits, the influence of post-crystallisation metamorphic fluids on the sulphide composition of the South Harris intrusives cannot be discounted. However, since sulphides remain relatively stable during metamorphic processes [37], and there is no clear petrographic evidence to suggest that substantial metamorphic alteration of the sulphides has occurred, it is unlikely that metamorphic fluids have significantly altered the sulphide composition of the South Harris intrusions. The undeformed, cubic nature of the sulphides in the South Harris intrusions (Figure 5B) are further indicative that these formed during magmatic crystallisation, though the sulphide veining may be indicative of minor metamorphic alterations.

The $\delta^{13}C$ ratios for the Bin Quarry graphitic pyroxenites (Table 3) are comparable to that of the Dalradian pelites, supporting a similar host rock carbon source for this locality [16]. The presence of abundant sulphides in this intrusion, with an average S content of 6.49% (Table 2), is also indicative of assimilation, with sulphur becoming incorporated into the melt during magmatic emplacement from the sulphidic Dalradian pelites. The ubiquitous, intracrystalline occurrence of sulphides and graphite throughout this deposit (Figure 6), including within the large bytownite (plagioclase) phenocrysts, suggests substantial early assimilation of the carbonaceous country rock prior to significant cooling and crystallisation. By contrast, sulphides and graphite within the South Harris metagranite

and metagabbro are inter-crystalline in nature (Figure 5), indicating crustal assimilation occurred at a late stage of magmatic emplacement and cooling.

### 4.2. Elemental Enrichment during Magmatic Assimilation

The minor enrichments in Cu, Mn, Se, Te, Ti, V, W and Zn in the South Harris intrusions relative to the upper crustal averages (Table 1; Figure 7) can confidently be associated with the assimilation of relatively enriched carbonaceous material. Black shales and their metamorphic derivates are commonly enriched in these redox-sensitive elements, where they are predominantly concentrated in sulphides and organic matter [2,38–41]. None of the enrichments in the South Harris intrusions are significant enough to be considered ore-grade, but their increased concentrations indicate a mobility of trace elements in the crust during magmatic emplacement. Trace element concentrations are broadly higher in the metagabbro than in the metagranite, with Cu, Mn, Ni, Te and Zn being 2× more abundant in the metagabbro (Figure 6), despite the average TOC concentrations being >5× higher in the metagranites. This may indicate a preferential affinity for trace element enrichment in mafic intrusions, or it may relate to the timing and conditions of crustal assimilation.

Trace element concentrations in the Bin Quarry graphitic pyroxenites are significantly higher than those found in the South Harris intrusions (Figure 6). Concentrations of Co, Cu, Ni, Se, Te and W are at least 10× those of the average upper crust [30–32], while Ag, Mo and Pd are at least 2× the upper crustal averages (Table 1 and Figure 7). This corresponds to significantly higher average concentrations for TOC (4.94%) and S (6.49%) in the Bin Quarry intrusions relative to the South Harris intrusions (Table 2), with elemental concentrations closely correlating with both TOC and S [16].

When compared to the trace element concentrations of the non-graphitic, layered olivine gabbros at Bin Quarry (Table 1; Figure 7), the graphitic pyroxenites demonstrate substantial trace element enrichments, particularly in Co, Cu, Ni and Pb, which are at least 10× more abundant. This increased concentration of redox-sensitive trace elements and their correlation with TOC and S in the Bin Quarry intrusion further support a model of crustal assimilation and element fixing from carbonaceous host lithologies. Trace element data from across the carbonaceous Dalradian metasediments in Scotland and Ireland [3] show these strata to be enriched relative to the average crustal and average shale values, with elevated semi-metal concentrations commonly associated with sulphides [42].

### 4.3. Implications for Orogenic Mineralisation

The Paleoproterozoic schists of the Outer Hebrides and the Neoproterozoic pelites of the Dalradian Supergroup are representative sedimentary deposits of the two most substantial periods of black shale deposition in the rock record [15] (Figure 1). Both sequences therefore represent a large sink of redox-sensitive elements in the upper crust, with the potential to act as significant sources of metals and semi-metals during magmatic assimilation.

Comparing the whole-rock and petrographic evidence from these two intrusive localities, there is evidence that the timing of crustal assimilation and the conditions of emplacement may be important controls of the extent of any elemental enrichment. Graphite and sulphides within the Bin Quarry intrusion are clearly abundant and homogeneously distributed throughout the intrusion, including within the plagioclase phenocrysts (Figure 6). This suggests that assimilation of the carbonaceous country rock occurred at depth prior to significant cooling and crystallisation of the magma. Conversely, the distribution of graphite and sulphides within the South Harris intrusions is less ubiquitous, with graphite and sulphides present as larger inter-crystalline minerals and veins (Figure 5). Most of the phenocrysts (orthoclase, amphibole, garnet and pyroxene) do not contain graphite or sulphides, except within fractures. This suggests an initial period of cooling, followed by the late-stage assimilation of carbonaceous country rock at shallower depths. This difference in paragenesis between these two intrusions may have affected the trace element mobilisation and fixing.

The contrasting petrographic and elemental enrichments identified in these two deposits could have additionally been influenced by variations in the redox state and sulphide saturation of the magmas during emplacement and crustal assimilation [43]. The rapid assimilation of sulphide-rich, reduced sedimentary lithologies within the Dalradian Supergroup may have resulted in substantial redox changes in the Ordovician Bin Quarry magma, causing the formation of localised, sulphide-rich graphitic pyroxenites (Figure 4), where the melt reached sulphide saturation. Slower assimilation at shallower depths in the South Harris magma may have caused sulphides to form later in the crystallisation process in less discrete bands. Variations in redox geochemistry and the timing of sulphide saturation in these melts may have also influenced the trace element enrichments of the resultant deposits [43].

Supracrustal rocks of the Paleoproterozoic and Neoproterozoic ages should be considered to have a high potential for base metal and other redox-sensitive element mineralisation, particularly where there was later magmatic emplacement. Although gabbros and granites are deeply crustally sourced, they are commonly graphitic and sulphidic; therefore, the application of the C-isotope analysis combined with S-isotopes can provide a method for determining carbon and sulphur provenance and identify any potential for mineralisation. A peak in the production of S-type granites in the Nuna supercontinent from 1.95 to 1.65 Ga [44] engendered a high degree of assimilation from Paleoproterozoic sedimentary rocks. Combined with the anomalously high TOC in sediments at that time, there was a strong opportunity for graphite-enhanced mineralisation in the plutons. For example, the mineralisation in graphite-bearing plutons includes base metals, gold and tin in Central Brazil [45] and tin, gold and PGEs in the Northern Territory/Queensland, Australia [46]. As well as substantial black shale deposits, the Paleoproterozoic and Neoproterozoic periods were notable for their abundance of volcanogenic massive sulphide (VMS) deposits globally, including examples from Gairloch (Paleoproterozoic) [36] and Creag Bhocan (Neoproterozoic) in Scotland [47]. Global VMS examples include the Palmeiropolis deposit in Brazil (Paleoproterozoic) [48], the Flin Flon and Flambeau deposits in North America (Paleoproterozoic) [49,50] and the Arabian-Nubian Shield of Northeast Africa (Neoproterozoic) [51]. VMS deposits could provide another more significant source for metal and semi-metal enrichment of magmatic intrusions during crustal assimilation. Targeted exploration for plutons that have interacted with VMS and/or highly carbonaceous lithologies at depth could help identify new metal deposits.

## 5. Conclusions

There is clear evidence of crustal assimilation of carbonaceous sedimentary lithologies in both the Paleoproterozoic intrusions of South Harris and the Neoproterozoic Bin Quarry intrusion of NE Scotland. The new carbon and sulphur isotopic analyses presented here demonstrate that the proximal Paleoproterozoic lithologies at Rodel, Stuaidh and Borve are the probable sources for the assimilated carbon and sulphur in the South Harris metagranite and metagabbros. A comparison with $\delta^{13}$C data from the Bin Quarry intrusion indicates that similar proximal crustal assimilation occurred in both lithologies, resulting in graphite and sulphide crystallisation. Trace elements were mobilised from the nearby carbonaceous sedimentary lithologies during magmatic assimilation and recrystallised within these plutons. A comparison of the whole-rock and petrographic analyses from the Bin Quarry intrusion with the South Harris plutons indicates that differences in the timing and depth of assimilation may affect the concentration and deportment of graphite, sulphides and trace elements in the plutons. Earlier crustal assimilation at greater depths may be favourable for greater assimilation and trace element mobilisation into plutons. Plutons hosted within Paleoproterozoic and Neoproterozoic carbonaceous deposits should be given significant consideration during the exploration for redox-sensitive trace elements. Given their laterally extensive nature and, therefore, the potential for the large-scale mineralisation of plutons during orogenic processes, sedimentary basins from these periods may host significant mineral deposits.

**Author Contributions:** Conceptualisation, J.G.T.A. and J.P.; methodology, J.G.T.A., J.P. and A.J.B.; validation, J.G.T.A., J.P and A.J.B.; formal analysis, J.G.T.A. and A.J.B.; investigation, J.G.T.A. and J.P.; resources, J.P. and A.J.B.; data curation, J.G.T.A., J.P. and A.J.B.; writing—original draft preparation, J.G.T.A.; writing—review and editing, J.P.; visualisation, J.G.T.A. and J.P.; project administration, J.P. and A.J.B. and funding acquisition, J.P. and A.J.B. All authors have read and agreed to the published version of the manuscript.

**Funding:** This research was funded by Natural Environment Research Council (NERC), grant NE/T003677/1.

**Data Availability Statement:** Not applicable. All relevant data are reported within the manuscript.

**Acknowledgments:** Thanks are given to Alison McDonald, John Still and Jason Donald for their skilled technical support.

**Conflicts of Interest:** The authors declare no conflict of interest.

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
