# Peer review of "Carbon in Mineralised Plutons"

_geosciences, doi:10.3390/geosciences12050202_

Round 1
Reviewer 1 Report
This is an interesting and novel work. The figures and dataset are mostly of high quality. I agree with most of the conclusions proposed by the authors. The writing and structure of the manuscript are well-prepared. Therefore, I recommend a minor revision. I have minor concerns and suggestions as listed below:
- Are the graphite and sulfides homogeneously present in the entire intrusion, or only present in local regions like the margin of the intrusion? Please clarify such information in the result section.
- The studied rocks are all metamorphic igneous rocks, how to preclude the influences of metamorphic fluids on the formation of sulfides?
- It is better to compare the metal and semi-metal concentrations (i.e., Cu, Mn, Se, Te, Ti, V, W and Zn) within the assimilated igneous rocks and unassimilated rocks to see whether the assimilated rocks really enriched in these elements. Comparison of the Cu, Ni elements in the pyroxenite with the average upper crust is meaningless, because ultramafic rocks generally contain higher concentration in terms of these elements. As a result, it is not convincing to indicate the assimilation process caused the enrichment of the mentioned metal elements.
- Pyrite and other sulfides are not common in granitic samples in comparison with magnetite. Crystallization of the sulfide minerals after magma assimilation process may indicate the variation of magma redox. Such issues could be further discussed in the manuscript.
Author Response
- Are the graphite and sulfides homogeneously present in the entire intrusion, or only present in local regions like the margin of the intrusion? Please clarify such information in the result section.
Graphite and sulphide contents of the intrusions is heterogeneous across the intrusions. An additional sub-section (3.1 Field Observations and Sampling) in the Results section has been added to provide this additional information requested.
- The studied rocks are all metamorphic igneous rocks, how to preclude the influences of metamorphic fluids on the formation of sulfides?
The s-isotope data discussed in the manuscript closely match a number of potential sulphur sources nearby the South Harris intrusions which may have been assimilated, indicating a proximal crustal source for sulphur. This is combined with the C-isotope data which also closely matches proximal schists, indicating assimilation was an active mechanism during the formation of these intrusions. Combined with the petrographic evidence presented, there is limited evidence of metamorphic alteration of the sulphides in the South Harris intrusions. The minor sulphide veining observed may indicate a degree of later fluid alteration. This has now been discussed in the text at length. The ubiquitous, intracrystalline nature of the sulphides in the Bin Quarry intrusion strongly indicates that these formed during magma crystallisation, rather than from metamorphic fluids. This is also discussed in the text.
- It is better to compare the metal and semi-metal concentrations (i.e., Cu, Mn, Se, Te, Ti, V, Wand Zn) within the assimilated igneous rocks and unassimilated rocks to see whether the assimilated rocks really enriched in these elements. Comparison of the Cu, Ni elements in the pyroxenite with the average upper crust is meaningless, because ultramafic rocks generally contain higher concentration in terms of these elements. As a result, it is not convincing to indicate the assimilation process caused the enrichment of the mentioned metal elements.
Additional analyses of the associated (non-graphitic) olivine gabbro from Bin Quarry has now been provided in the data table and discussed in the manuscript. It is shown that elemental concentrations (including Cu & Ni) at both Bin Quarry and South Harris are elevated in comparison to the olivine gabbro, indicating an additional crustal source for these elements. Elemental data for the unassimilated rocks is not available for comparison, though several references are now provided to indicate the semi-metal enriched nature of carbonaceous sedimentary rocks, including the Dalradian metasediments specifically.
- Pyrite and other sulfides are not common in granitic samples in comparison with magnetite. Crystallization of the sulfide minerals after magma assimilation process may indicate the variation of magma redox. Such issues could be further discussed in the manuscript.
This is true. Redox changes due to assimilation of carbonaceous country rock as well as sulphide saturation of the melt by assimilation of sulphidic sediments are both possible mechanisms for causing sulphide crystallisation in these intrusions, including the granites. This could also explain some of the trace element concentration differences between the two deposits This has now been discussed in the text.
Reviewer 2 Report
This is well-written, well-structured and concise article dealing with the origin of graphite within gabbroic and granitic plutons. Authors employed whole-rock elemental, total organic carbon and sulphur concentrations combined with carbon and sulphur isotope analyses of graphite and sulphides to document the assimilation of carbonaceous lithologies during various stages of magmatic pluton emplacement. Based on intra-crystalline occurrence of graphite and sulphides, authors suggest early assimilation of country rocks in pyroxenites, contrasting with late-stage crustal assimilation in metagranite and metagabbro documented by BSE images of inter-crystalline graphite and sulphides. These inferences could be strengthened by temperatures calculated from vibrational spectroscopic characteristics of graphite calibrated up to 780 °C (e.g., Hurai et al. 2015, Geofluids, Elsevier, p. 253).
There are some inconsistencies which should be clarified and fixed before acceptation:
Lines 2, 11: Is the term “mineralized, mineralization” consistent with the British spelling?
Lines 46, 47: missing logic
Lines 67-84: whole paragraph looks like Discussion rather than Geological Setting
Figures 4 and 5: use IMA-approved abbreviations of minerals (Warr 2021, Mineralogical Magazine)
Tables 1-4: format inconsistent with editorial rules: centred cells, do not use vertical dividing lines, use reference numbers instead author names, etc.
Table 1 should be rearranged according to intsructions in annotated pdf
Whole manuscript: use en-dash, not hyphen, to express range of numbers
Additional minor spelling errors and flaws are highlighted in the annotated pdf

Author Response
These inferences could be strengthened by temperatures calculated from vibrational spectroscopic characteristics of graphite calibrated up to 780 °C (e.g.,Hurai et al. 2015, Geofluids, Elsevier, p. 253).
While the authors agree that the work may be strengthened by raman spectroscopy, this is out-with the analytical scope of this work. Additionally, based on the evidence and conclusions of this work, (that graphite at both localities was magmatically assimilated, early vs late), raman spectroscopy would likely return an above maximum temperature (>780 °C) value from graphite at both localities and therefore the additional benefit would likely be limited.
There are some inconsistencies which should be clarified and fixed before acceptation:
Lines 2, 11: Is the term “mineralized, mineralization” consistent with the British spelling?
'Mineralized' has been used throughout the manuscript to follow the convention set by the title of this topical collection (‘Geological Features on Magmatic–Hydrothermal Mineralization’). There is no clear British vs US spelling regarding 'mineralised' or 'mineralized'. According to the Geological Society of London, both spellings are acceptable. No change made.
Lines 46, 47: missing logic
This paragraph has been re-worded to make meaning clear.
Lines 67-84: whole paragraph looks like Discussion rather than Geological Setting
Lines 79-84 have been removed as these were more discussion than geological setting and similar statements are made again in the discussion anyway. Lines 67-79 have remained in the manuscript, with minor edits, as they lay out the existing geological understanding of the two areas.
Figures 4 and 5: use IMA-approved abbreviations of minerals (Warr 2021, Mineralogical Magazine)
Figures 4 & 5 have been altered to utilise Warr, 2021 abbreviations, as suggested.
Tables 1-4: format inconsistent with editorial rules: centred cells, do not use vertical dividing lines, use reference numbers instead author names, etc.
All tables now formatted as per editorial rules. Table 1 merged as indicated, though this has resulted in Table 1 being inserted in a landscape orientation, with a reduced font size. Please advise if this is unsuitable.
Table 1 should be rearranged according to intsructions in annotated pdf
As above – Table 1 rearranged in landscape with smaller text through necessity. Please advise if unsuitable.
Whole manuscript: use en-dash, not hyphen, to express range of numbers
This has been amended throughout the manuscript
Additional minor spelling errors and flaws are highlighted in the annotated pdf
All indicated errors highlighted have been amended (see comments in attached pdf)
